# Diabetic Retinopathy and Diabetic Macular Edema Detection Using Ensemble Based Convolutional Neural Networks

**DOI:** 10.3390/diagnostics13051001

**Published:** 2023-03-06

**Authors:** Swaminathan Sundaram, Meganathan Selvamani, Sekar Kidambi Raju, Seethalakshmi Ramaswamy, Saiful Islam, Jae-Hyuk Cha, Nouf Abdullah Almujally, Ahmed Elaraby

**Affiliations:** 1Department of CSE, SASTRA Deemed University, SRC Kumbakonam, Thanjavur 612001, India; 2School of Computing, SASTRA Deemed University, Thanjavur 613401, India; 3Department of Maths, School of SASH, SASTRA Deemed University, Thanjavur 613401, India; 4College of Engineering, King Khalid University, Abha 61421, Saudi Arabia; 5Department of Computer Science, Hanyang University, Seoul 04763, Republic of Korea; 6Department of Information Systems, College of Computer and Information Sciences, Princess Nourah Bint Abdulrahman University, Riyadh 11671, Saudi Arabia; 7Department of Computer Science, Faculty of Computer Science and Information, South Valley University, Qena 83523, Egypt; 8Department of Cybersecurity, College of Engineering and Information Technology, Buraydah Private Colleges, Buraydah 51418, Saudi Arabia

**Keywords:** diabetic retinopathy, ensemble convolutional neural network, diabetic macular edema, Harris hawks optimization and artificial intelligence

## Abstract

Diabetic retinopathy (DR) and diabetic macular edema (DME) are forms of eye illness caused by diabetes that affects the blood vessels in the eyes, with the ground occupied by lesions of varied extent determining the disease burden. This is among the most common cause of visual impairment in the working population. Various factors have been discovered to play an important role in a person’s growth of this condition. Among the essential elements at the top of the list are anxiety and long-term diabetes. If not detected early, this illness might result in permanent eyesight loss. The damage can be reduced or avoided if it is recognized ahead of time. Unfortunately, due to the time and arduous nature of the diagnosing process, it is harder to identify the prevalence of this condition. Skilled doctors manually review digital color images to look for damage produced by vascular anomalies, the most common complication of diabetic retinopathy. Even though this procedure is reasonably accurate, it is quite pricey. The delays highlight the necessity for diagnosis to be automated, which will have a considerable positive significant impact on the health sector. The use of AI in diagnosing the disease has yielded promising and dependable findings in recent years, which is the impetus for this publication. This article used ensemble convolutional neural network (ECNN) to diagnose DR and DME automatically, with accurate results of 99 percent. This result was achieved using preprocessing, blood vessel segmentation, feature extraction, and classification. For contrast enhancement, the Harris hawks optimization (HHO) technique is presented. Finally, the experiments were conducted for two kinds of datasets: IDRiR and Messidor for accuracy, precision, recall, F-score, computational time, and error rate.

## 1. Introduction

Computer-assisted health care, health care technology consulting, and health monitoring equipment are just a few of the current buzz words. Thanks to the connection and computing architecture that has drawn attention to the electronic era we live in, ordinary people now have the luxury of receiving diagnosis and treatment from the comforts of home with a single tap [1,2,3]. While routine illnesses and minor illnesses can usually be treated without visiting a doctor, some more severe illnesses still necessitate a great deal of effort from the medical establishment. Technology can help, but not replace human intervention. With the advancement in AI technology, technologies can now autonomously analyze a patient’s condition and identify a condition in a matter of seconds using the patient’s significant history and associated data [4,5,6]. By 2025, the amount of DR individuals suffering is predicted to rise from 382 million to 592 million. According to a study conducted in the Pakistani province of Khyber Pakhtunkhwa (KPK), 30 percent of diabetic individuals suffer from DR, with 5.6 percent going blind [7,8,9]. If mild NPDR is not treated in the beginning phases, it might progress to PDR. In another study, 130 people with DR symptoms were found in Sindh, Pakistan [10,11]. According to the findings, DR patients made up 23.85% of the overall examined patients, with PDR patients accounting for 25.8% [12,13]. Patients with DR are symptomatic in the beginning phases; however, as the disease progresses, it causes blobs, vision problems, distortions, and gradual visual acuity loss. Diabetic retinopathy is one of the issues previously mentioned in the article. Diabetic retinopathy is caused by diabetes destroying the blood flow on the retina’s inner, resulting in blood and other body fluids leaking into the tissues surrounding it. Soft, damaged tissue (also known as cotton wool patches) [14], hard exudates, microaneurysms, and hemorrhages form as little more than a result of the leaking [15]. It is the most common cause of visual loss in the working-age population [16]. Diabetic retinopathy (DR) is caused due to diabetes mellitus, which can damage the retina and even lead to the loss of vision. The DR has several stages of severity such as mild, moderate, and severe [17]. The severe stage of DR is termed as proliferative diabetic retinopathy (PDR), in which the formation of new vessels in the retina is observed [18]. However, the early detection of DR and proper diagnosis will reverse or reduce the growth of the effects caused by the disease. Diabetic macular edema (DME) is a condition in which the lesions caused by DR are observed in the middle portion of the retina called the macula. The DME is considered as a serious condition as the damage caused by it is irreversible. The identification of features such as micro-aneurysms, hard exudates, hemorrhages, etc., can be used to carry out the detection of these diseases. These micro-aneurysms refer to the red spots in the retina’s blood vessels with sharp margins formed in the early stages of the disease. The exudates are caused due to abnormality in the blood vessels, which are formed as yellowish-white spots in the outer layer of the retina. Hemorrhages also occur such as micro-aneurysms but have irregular margins caused due to the leakage of capillaries. The blockage of arteries also contributes to cotton wool spots, which occur as a white region in the retinal nerve. Several methods have been developed for the detection of DR and DME to provide diagnosis, but these traditional methods were inefficient in accurately detecting diseases. Deep learning techniques have been deployed for disease detection in which the retinal image (fundus image) is used as the input in which the features are extracted for detection. These approaches have been found to be more effective in identifying features than the traditional methods; however, these approaches also suffer from inaccuracy due to the presence of noises and artifacts in the input images.

Figure 1 describes the retina images for disease DR and DME. As a result, it is hard but critical to recognize DR to prevent the worst effects of later stages. Fundus imaging is utilized to diagnose DR, as mentioned in the preceding section. Manual analysis can only be performed by highly qualified subject matter experts and is thus cost and time intensive. As a result, it is critical to apply machine vision technologies to assess the retina image features and aid physicians and radiologists. Hands-on development and end-to-end learning are two types of computer vision-based methodologies. Traditional algorithms such as HoG, SIFT, LBP, Gaussian filters, and others are used to extract the features; however, they failed to preserve the scale, rotation, and brightness fluctuations [19].

Several existing approaches have integrated the preprocessing of input images and the deep learning-based detection of diseases in which the accuracy in the detection of diseases was observed to be improved. The common processes involved in these approaches are the preprocessing of input images, enhancement in contrast, and the extraction of features for the detection of diseases. The machine learning models such as support vector machine (SVM) and K-nearest neighbor (KNN) classifiers were found to be appropriate for detecting DR and DME. The severity of the disease was determined by the number of features identified by the model; however, the imbalance in the distribution of datasets resulted in the inefficient determination of severity. In particular, an effective mechanism in the detection of DR and DME, along with the determination of severity, is still in demand. The major aim of this research work was to provide the effective detection of DR and DME and to determine the disease’s severity to define the disease’s damage level on the patient. The accuracy of detection was achieved by performing the proper processing of the input retinal image. End-to-end learning understands the underlying rich traits dynamically, allowing for greater identification. Inside the retina imaging databases, many hand-on engineering and end-to-end learning-based algorithms have been used to identify the DR. Still, none of them can identify the mild stage. Accurate diagnosis of the weak stage is critical for controlling this devastating disease. Utilizing end-to-end deep ensembles models, this study attempted to discover all stages of DR (including the moderate stage). The findings revealed that the proposed strategy beats the current methods.

The major objective of this research is to provide precise classification between the DR and DME and to compute the severity of the diseases accurately. This objective can be achieved by fulfilling the sub-objectives, which are listed as follows,
To minimize the noise level in the input image by performing effective preprocessing of the image;To maximize the precise identification of features from the preprocessed image by enhancing the contrast level;To maximize the accuracy of detection by incorporating the segmentation of lesions in the blood vessels;Effectively classify the images into three classes based on the extraction of significant features;To determine the severity of the disease based on the variation in the intensity of the features for diagnosis.

The major contributions of this paper are as follows:
In our work, we performed preprocessing that included three processes such as noise removal using iterative expectation maximization, artifact removal using nonlinear filtering, and contrast enhancement using Harris hawks optimization; the preprocessed image was used to enhance the quality of the images, which led to high segmentation and detection accuracy. Preprocessing was performed to reduce noise and artifacts and improve the contrast, which increased the efficiency of feature extraction and reduced the false detection rate.Segmentation was performed before feature extraction and classification, which increased the detection accuracy. For segmentation, we proposed improved OPTICS clustering, which considers particular regions of interest and takes less time for segmentation, thus reducing latency and increasing the disease detection accuracy.Improved OPTICS clustering overcomes misalignment problems due to considering the particular region of interest, thus increasing the segmentation and detection accuracy.The extraction of features was carried out in the segmented images obtained from the previous process. Features such as micro-aneurysms, hemorrhages, and hard exudates, collectively termed as structural features, are considered the essential features; along with this, the shape features, orientation features, and color features are also considered for the classification of DR and DME. The ensemble CNN architecture was implemented for this purpose, which outperformed the ensemble CNN class prediction. From this, the classification of images was carried out in several classes, namely, normal, DR, and DME. Furthermore, the severity of the disease was computed by using conditional entropy in which the number of lesions is considered for the threshold generation. Based on the threshold, the severity level of the disease was classified into three classes: mid, moderate, and severe.The proposed research work is evaluated in terms of performance metrics such as accuracy, precision, recall, F-score, computation time, and error rate.

The rest of the paper is organized as follows: Section 2 illustrates the state-of-the-art in diabetic retinopathy and diabetic macular edema detection using specific approaches. Section 3 discusses the major problems that exist in this field. Section 4 describes the system model with the proposed algorithms and techniques in detail. Section 5 describes the experimental results of the proposed as well as previous methods. Section 6 concludes the paper by providing future enhancements.

## 2. Related Work

In the literature, the diagnosis of DR has received much interest. In [20], researchers offered a robust system that automatically recognized and classified retinal lesions (blood vessels, microaneurysms, and exudates) from retinal imaging. Blood vessels, microaneurysms, and exudates were first discovered using image processing methods. Following this, the retina properties of the vascular system, microaneurysm count, exudate area, contrast, and homogenization were evaluated from the images obtained. These characteristics were then fed into a fuzzy classifier that uses the information to classify healthy, mild NPDR, moderate NPDR, severe NPDR, and PDR stages. A sample of 40 color fundus images was obtained from the DIARETDB0, DIARETDB1, and STARE datasets using a fuzzy classifier, correctly classifying the images with an efficiency of up to 95.63 percent.

A reliable automated approach for detecting and classifying the various stages of DR has been suggested The optic disc and retina neurons are separated, and characteristics are retrieved using the gray level co-occurrence matrix (GLCM) approach. To identify various stages of DR, a fuzzy classifier and a convolutional neural network were used to classify them. DIARETDB0, STARE, and DIARETDB1 were the datasets used [21].

The unique clustering-based automatic region growth methodology was introduced in this study. Several types of features—waveform (W), co-occurrence matrix (COM), histogram (H), and run-length matrix (RLM)—were retrieved for the texture features, and several ML algorithms were used to achieve a classification performance of 77.67 percent, 80 percent, 89.87 percent, and 96.33 percent, respectively. The information fusion approach was utilized to create a fused hybrid-feature database to improve the accuracy of the classification. Two hundred and forty-five elements of the hybrids’ feature data (H, W, COM, and RLM) were extracted from each image, and 13 optimum characteristics were chosen using four methodologies: Fischer, mutual information feature selection, information gain, and the possibility of the dependent variable average correlation [22]. The number of DR patients outnumbered the number of practitioners by a large margin. As a result, manual clinical diagnosis or screening takes a long time. To avoid this problem, follow-up scanning is performed regularly, and automated DR identification and intensity classification are required. Several strategies for detecting retinopathy and classifying its severity and likelihood are presented here [23].

Exudates are the diagnostic indications of diabetic retinopathy, a retina condition caused by long-term diabetes that can lead to eyesight problems if not detected early. The procedure of recognizing and categorizing exudates from a retinal image has been made easier thanks to a medical screening program. The exudates are first segregated using the FCM technique and then transformed into discrete mother wavelets. The classifier is fed the texture textural properties retrieved by the grey-level co-occurrence matrix. The suggested program’s efficiency was evaluated by comparing it to the data from the publicly available dataset IDRID. MATLAB was used to formulate and construct a GUI [24].

This research has the proposed texture feature extraction characteristics of the GLDM method (contrast, angular second moments, density, median, and inverse difference moment) feature and feed-forward neural net classifier as a machine learning-based approach for DR detection and evaluation. According to the results of the trials and performance assessment, the suggested methodology had a detection performance of 95% [25].

Diabetes is responsible for 50 deaths per 1000 live births amongst individuals over the age of 70. The identification of diabetes at a preliminary phase and the implementation of a suitable therapy may minimize the visual loss among the sufferers. Once symptoms of DR have been identified, the severity of the disease must be defined to recommend the appropriate treatment. Mild nonproliferative diabetic retinopathy (NPDR), moderate NPDR, severe NPDR, proliferative diabetic retinopathy (PDR), and no DR are the five phases of diabetic retinopathy severity. The techniques and issues associated with DR identification are summarized in this publication [26]. In [27], the authors proposed diabetic retinopathy classification using retinal images through an ensemble learning algorithm. The proposed work includes the following processes: retinal image collection, preprocessing, feature extraction, and feature selection and classification. In preprocessing, the noisy images, duplicate images, and black borders are removed from the images. Tone mapping is used to increase the contrast and luminance in the images. Two sets of features are extracted from images such as the histogram-based feature and GLCM feature extraction. Then, the features are concatenated to select the relevant features. Here, the GA algorithm is used for feature selection. Finally, classification was undertaken by the XGBoost algorithm using the selected features. Here, genetic algorithm (GA) was used for feature selection; it takes a lot of time to select the features, thus increasing feature selection and classification latency.

Early detection of diabetic retinopathy using retinal images for diabetes is presented in [28]. The proposed method includes four processes: preprocessing, segmentation, feature extraction, and classification. The preprocessing includes noise removal and contrast enhancement using histogram equalization (HE). The segmentation is performed by Gaussian derivative and Coye filter, which segments the EX, MA, and HM. The features are extracted from the segmented image and extract features such as EX, MA, and HM values. Finally, SVM is used to classify the images using the extracted features. Here, SVM was used for classification, which takes a lot of time for training when considering larger datasets, thus leading to classification latency.

A histogram equalization method for the early detection of diabetic retinopathy was presented in [29]. The proposed algorithm included three methods: histogram clipping, RIHE-RVE, and RIHE-RRVE, which addressed the issues of the illumination of the retinal images. To avoid enhancement, the histogram clipping algorithm was proposed. The simulation result showed that the proposed method achieved a high performance compared to the other state-of-the-art methods. Here, the histogram equalization method was proposed, however, it is an unselective process that may increase the background noise contrast while decreasing the functional input image.

The authors in [30] proposed CANet to detect diabetic retinopathy and macular edema for diabetes. The proposed work used ResNet50 to produce a feature map with various resolutions including a cross-disease attention network, disease-specific attention module, and disease-dependent attention module. The disease-specific attention module was used to learn the features of the two diseases. In this stage, the inter special relationship was evaluated to detect the diseases. A disease dependent attention module was used to evaluate the internal relationship between the DR and DME diseases. Here, raw images were considered for training and testing, thus increasing the high false positive rate due to the presence of noise and low contrast, also reducing the detection accuracy.

The authors in [31] proposed a deep learning algorithm to detect diabetic retinopathy disease in diabetic patients. The proposed method included two processes: diagnosing DR severity and the feature extraction of DR. The proposed system hierarchical multitask learning architecture aims to detect both the DR severity and DR feature extraction. Finally, the fully connected layer provides the output, and it considers the hybrid loss, cross-entropy loss, and kappa loss for reducing the errors in the levels of DR severity. The simulation results showed that the proposed model achieved a higher performance using traditional deep learning methods. Here, the traditional deep learning method was used to detect the DR severity levels and feature extraction of DR; however, it generated multiple convolutional layers, thus increasing the complexity and latency.

In [32], the authors proposed a modified contrast enhancement approach from the effective identification of features in detecting diabetic retinopathy and diabetic macular edema. The limitations of conventional contrast limited the adaptive histogram equalization (CLAHE) technique such as the fixed clip limit and region of context, resulting in the inefficient identification of minute features, but can be overcome by implementing modified particle swarm optimization (MPSO) to determine the optimal clip limit and region of context, thereby resulting in the precise identification of features that further help in the accurate detection of diseases. The global best solution of all the operating particles was computed by comparing the output provided by all the particles in the iteration, which resulted in enhanced image contrast. The optimization of the clip limit and region of context was performed by the MPSO algorithm for the purpose of enhancing the contrast of the input image, but the proposed algorithm possessed slow convergence and is stuck in the optimal local solution.

In [33], the authors proposed an approach for the detection of diabetic macular edema in an automatic manner. The macular edema was identified, and the severity of the disease was determined by implementing mathematical morphology. The retinal image was used as the input from the detection process that was carried out. Initially, the preprocessing of the input image was performed from the removal of noise and enhancement of the contrast. Furthermore, the localization of the macula was executed by removing the optic disc and locating the center of the fovea. Then, the exudates in the region of the macula were identified in order to determine the severity of the disease. The removal of artifacts such as reflection due to lighting was removed as a post-processing step to achieve an accurate determination of severity. The detection of the macula in the input retinal image was carried out by using mathematical morphology, but this approach resulted in less accuracy in the detection of the macula region.

In [34], the authors proposed a probability-based construction of the future retinal image in detecting diabetic retinopathy. The difficulty in identifying the future instances of lesions in the retinal image was addressed. Initially, the segmentation of lesions and vessels was carried out to identify the severity of the disease from the input retinal image. Then, the probability of future lesion location was computed by the construction of a probability map. Furthermore, the generated probability map, along with the structure of vessels, was considered for the systemization of future lesions in the retina. This method was found to be effective in predicting future lesions based on the progression of the severity of diseases. The future severity of diabetic retinopathy was determined by using the probability map and the features of the current vessels, but the lack of noise removal in the input image reduced the efficiency of this approach.

## 3. Problem Statement

An input fundus image is used to perform the identification of diabetic retinopathy and diabetic macular edema; however, the accuracy of the system is decreased by the increased false detection rate of the existing techniques. In addition, the following issues are encountered in the best detection of DR and DME, which are listed as:Difficulty in feature differentiation: The detection of DR and DME is based on various features such as hard exudates, hemorrhages, and micro-aneurysms, but the differentiation of these minute features from each other is a hard task, which degrades the computation of the accurate severity of diseases.Class Overlapping: Current techniques also consider illness severity; however, the sparse training data for each severity leads to class imbalance issues that degrade the classification accuracy.Inadequate preprocessing: Using the current methods for effective contrast enhancement with traditional preprocessing leads to difficulties distinguishing features from the background.

In [35], the authors proposed diabetic retinopathy detection using a deep convolution neural network (DCNN) for nonproliferative diabetic retinopathy. The proposed work includes three phases: preprocessing, candidate lesion detection, and candidate extraction. In preprocessing, the image contrast is enhanced using curve transformation. Then, the images are smoothened by a bandpass filter. In the lesion, the detection process includes four stages: optical disc removal, candidate lesion detection, vessel extraction, and preprocessing. In candidate extraction, the micro-aneurysms are detected to measure the coefficient between every pixel using Gaussian kernels. For this, a PCA algorithm was proposed to reduce the dimensionality. Finally, classification was undertaken by DCNN. In this way, the proposed work achieved high accuracy of nonproliferated diabetic retinopathy. The major issues determined in this paper are as follows:Here, preprocessing was performed to enhance the quality of the retinal images; however, the retina image still has noise due to the implementation of traditional contrast enhancement techniques, thus reducing the image quality, which leads to a high false detection and reduced detection accuracy.DCNN is used for feature extraction and the detection of nonproliferated diabetic retinopathy. Still, DCNN focuses on the whole image for the extraction of features without any particular region of interest, thus increasing the high latency for feature detection.The PCA algorithm was used to reduce the dimensionality, but the number of principal components must be selected otherwise it may cause information loss, thus reducing the detection accuracy.

The authors in [36] proposed a data augmentation method to improve the detection rate of proliferative diabetic retinopathy. The NVs were inserted onto pixels located on vessels. Vessel segmentation was performed by Otsu thresholding and the U-Net deep learning algorithm, and then optic disc segmentation was performed. The count of NVs was determined by selecting random values using a threshold. The next process is semi-random blood vessel generation, which is based on the tree structure. This process considers the shape and orientation of the NVs. For the vessel color assignment color, a matrix was proposed that calculates the weighted average of the RGB values of the images. Finally, DR grading and data augmentation was proposed to improve the NVs. Some of the significant problems in this research are as follows:Here, the Otsu thresholding method was used for vessel segmentation, which performed well; however, it did not provide an optimal result for noisy images. First, the noise is removed from the images, and then the thresholding is applied; otherwise, this method will fail, thus reducing the performance of vessel segmentation.The detection of diabetic retinopathy was carried out by performing the segmentation of neovessels in the retina. However, performing detection based on a single feature results in a high false detection rate.Here, the U-Net algorithm was also used for vessel segmentation, which takes a lot of time to learn the vessels from the retinal images at the middle layers, thus leading to high latency.

The authors in [37] proposed the analysis of retinal images to detect eye diseases for diabetes using the deep learning method. The proposed method considered two processes: detection and localization, and the segmentation of localized regions. For localization, the author proposed the FRCNN method, which extracts the features from the images that evaluate the affected portions. For the segmentation process, the author proposed the FKM clustering algorithm. The ground truth was generated for detecting the affected regions during training. Finally, the DME is classified into two classes such as DME and background. The serious issues in this paper are as follows:Here, raw images were considered for the localization and segmentation process, thus reducing segmentation and detection accuracy due to low contrast and the presence of noise in the retinal images.Faster RCNN was implemented for the extraction of features but the lack of pixel-to-pixel alignment in the region of interest caused misalignment, resulting in the degradation of the detection accuracy.The proposed approach was used for diabetic-based disease detection in the eye, but the detection of various diseases from the limited number of trained images resulted in class imbalances.

The authors in [38] presented an efficient framework for the detection of macular edema disease for diabetes. The proposed work used the combination of a deep convolution neural network (DCNN) and a meta-heuristic algorithm for feature extraction and feature selection, respectively. At the stage of feature extraction, the proposed work reduced the feature extraction complexity by reducing the prior knowledge. The SMOTE algorithm was used to perform class imbalance. The generic algorithm and binary particle swarm optimization algorithm were used to select the relevant features. The drawbacks in this paper are as follows:Here, the features were extracted from the noisy images, thus reducing the quality of the images and leading to poor feature extraction, thus increasing the macular edema’s false detection rate.The integration of the genetic algorithm and binary particle swarm optimization was used to determine the subset size. However, implementing these two algorithms increases the complexity and time consumption, thereby increasing the latency.DCNN was used for feature extraction and the detection of nonproliferated diabetic retinopathy, however, DCNN focuses on the whole image for the extraction of features without any particular region of interest, thus increasing the high latency for feature detection.

## 4. Proposed Model

In this research work, we concentrated on accurately detecting the DR and DME from the input fundus images. The severity of the disease is also determined based on the features extracted from the images. Figure 2 shows the architectural view of the proposed work. The description of the dataset is provided below:

The properties of the blood vessels in the retinal image enable the ophthalmologist to assess retinal disease. The presence of lesions on the fundus image is the first sign of diabetic retinopathy. The preprocessing technique is mainly used to remove unwanted noise and enhance some image features. 

The fundamental idea underlying OPTICS is to find the points associated by density to extract the cluster structure of a dataset. The approach generates a density-based representation of the data by constructing a reachability graph, an ordered collection of points. Each location in the list has a reachability distance associated with it, which measures how simple it is to get to that site from other points in the collection. Points with comparable accessibility distances are most likely in the same category.

Before sharing our preprocessed image with CNN, we converted the image to an array and mapped that array’s values in the range of 0 to 1 as the epoch was set at 235 to reach a deep network. The initial learning rate was kept at 1 × 10³, which is the default value for the Adam’s optimizer, and the die stack size was 32. We trained our model with more pictures, obtained only a few hundred of images for training, and generated more images from the existing dataset by passing parameters such as the rotating range, width changing range, height changing range, scissors range, zoom range, and pan on image data generator.

The classification of diabetic retinopathy is classified into two types: nonproliferative and proliferative. The term “proliferative” refers to whether the retina has neovascularization (abnormal blood vessel growth). Nonproliferative diabetic retinopathy refers to early illness without neovascularization (NPDR).

Dataset Collection: For accurate prediction of diabetic retinopathy and diabetic macular edema, we applied two kinds of retina fundus images: IDRiD and MESSIDOR. The description of these two datasets is as follows:IDRiD: Based on the presence of DR and DME disease, 516 images were loaded in the dataset. In addition, images were acquired through the field of view and stored in JPG format, and the size of each image was 800 KB. This dataset contained 81 color fundus images with the sign of DR. With this dataset, hard exudates (EX), microaneurysms (MA), soft exudates (SE), and hemorrhage (HE) based images are stored.MESSIDOR: This dataset was used, whose scope is to develop the DR and DME detection of images. In total, 1200 eye fundus images were used with the multiple pixel rates of 1440 × 960, 2240 × 1488, and 2304 × 1536.

The following steps implement a prediction of DR and DME.

### 4.1. Preprocessing

This is an initial step for DR and DME detection. To enhance the information for the disease diagnosis system, it is necessary to use some of the preprocessing steps as follows:

(a) Noise Filtering: Fundus images are cropped by salt and pepper noises, which are removed from the input images using the iterative expectation maximization (IEM) approach. In this approach, uncertainty is overwhelmed by using IEM variables. Noise is removed in the zig-zag trajectory and edge, and the corner position of the image is denoised using IEM variables. A dynamic threshold was computed and adjusted accordingly for noise removal since the acquisition of each image was different with their resolution.

The proposed inverse dual tree initial ranging (IDTIR) procedure uses the iterative expectation maximization (IEM) algorithm. The IEM algorithm is an iterative method that effectively estimates the parameters of the statistical model. In the IEM algorithm, two major steps are executed to estimate the parameters accurately. These steps can be explained as follows:E-step—This step determines the current estimate of parameters by creating a function for the expectation of log-likelihood. The expectation step is the base of the proposed IEM algorithm.M-step—This step is the final step that computes the parameters in such a way that the expected log-likelihood function can be maximized (i.e., the likelihood function determined in the E-step is maximized to calculate the parameters.

The above two steps were iteratively executed to determine the final parameters. Let $\omega_{V,L}$ be the parameter vector, and it can be represented as $\omega_{V,L}=\left[h_{V,L},T_{V,L}\right]\in \mu$ for the $L^{\mathrm{th}}$ active channel path of the given ranging code. The set of parameters is represented as μ. The latest estimated parameter set is denoted as $\hat{\mu }$ and can be formulated as follows,
(1)$${\hat{\omega }}_{V,L}=\left[{\hat{h}}_{V,L},{\hat{T}}_{V,L}\right]\in \hat{\mu }$$

E-Step Computation

In this step, the expected value is calculated as
(2)$$G\left(\omega_{V,L}|\hat{\mu }\right)≜\ln P\left({Y|\omega }_{V,L},\hat{\mu }\right)\alpha -\|\left(Y-{\hat{K}}_{V,L}\right)-h_{V,L}\Gamma \left(T_{V,L}\right){∁}_{V}\|{}^{2}$$

Here,
(3)$${\hat{K}}_{V,}≜\overset{N}{\underset{ℵ=1}{\sum }}\overset{N_{B}}{\underset{s=1}{\sum }}{\hat{h}}_{ℵ,s}\Gamma ({\hat{T}}_{ℵ,s}){∁}_{ℵ}-{\hat{h}}_{V,L}\Gamma \left({\hat{T}}_{V,L}\right){∁}_{V}$$

M-Step Computation

In this step, the expected value is maximized as follows:(4)$${\hat{\omega }}_{V,L}=\arg\max\left\{G\left(\omega_{V,L}|\hat{\mu }\right)\right\}$$

After parameter estimation, the estimated parameters are updated in the parameter vector. These two steps are executed until the terminating condition is met. The channel coefficient is derived from the parameter vector by letting the derivative equal zero with the fixed timing offset.

(b) Artifact Removal: Blurriness, poor edges, and illumination are called artifacts, which are removed using the nonlinear diffusion filtering algorithm, which eliminates all kinds of artifacts and ensures the image quality in terms of illumination correction and edge preservation.

(c) Contrast Enhancement: Low contrast is one of the important issues of image classification. In this work, we considered contrast enhancement as an optimization problem with the intention of optimizing the pixel values based on the contrast level of the input image. To enhance the contrast level of the input image, we proposed the Harris hawks optimization algorithm, which improves the performance of the image brightness.

H2O is a recently developed meta-heuristic algorithm that performs better in solving optimization problems. The H2O algorithm mimics the cooperative strategy and chasing style of the Harris hawks in nature. Since it has an intelligent searching strategy and fast convergence rate, it works better than the conventional genetic algorithm, particle swarm optimization algorithm, etc. Due to the benefits of the H2O algorithm, it was adapted for contrast enhancement using the pixel intensity rate in the proposed system. The proposed H2ORSS algorithm detects the optimum threshold value for replacing the pixel intensity values with normal ones. The proposed H2ORSS algorithm involves three major processes: initialization, fitness value estimation, and update of hawks.

Initially, the image matrix is initialized as hawks with the population size of PS. For each hawk ($X_{i}$) in the population, the fitness function is estimated. The fitness function is determined in terms of the pixel intensity, neighbor intensity, and resolution. The fitness function of the ith hawk is expressed as follows, Once the fitness is computed for all hawks, then three sequential phases are executed to select the optimal solution.

Phase 1: Exploration Phase

This phase relies on waiting, searching, and detecting prey. In every step, each Harris hawk is considered as the alternative solution. Based on the fittest solution, the position for each Harris hawk is updated as follows:(5)$$X\left(\mathrm{iter}+1\right)=\left\{\begin{array}{c} X_{\mathrm{rand}}\left(\mathrm{iter}\right)-{𝓻}_{1}\left|X_{\mathrm{rand}}\left(\mathrm{iter}\right)-2{𝓻}_{2}X\left(\mathrm{iter}\right)\right|\;\;\;\mathrm{if}\;𝓸\geq 0.5\\ X_{p}\left(\mathrm{iter}\right)-X_{a}\left(\mathrm{iter}\right)-{𝓻}_{3}\left(l_{b}+{𝓻}_{4}\left(u_{b}-l_{b}\right)\right)\;\;\;\mathrm{if}\;𝓸<0.5 \end{array}\right.$$

The location of the hawks in the next iteration is denoted as $X\left(\mathrm{iter}+1\right)$ and ${𝓻}_{1}$,$\;{𝓻}_{2}$, ${𝓻}_{3}$,$\;{𝓻}_{4}$ are the present location vectors of the hawks. Furthermore, 𝓸 is the random number selected in the range of 0 and 1, and $u_{b},l_{b}$ are the upper bound and lower bound, respectively. The average location of hawks ($X_{a}\left(\mathrm{iter}\right)$ can be estimated from the following expression:(6)$$X_{a}\left(\mathrm{iter}\right)=\frac{1}{\mathrm{PS}}\overset{\mathrm{PS}}{\underset{i=1}{\sum }}X_{i}\left(\mathrm{iter}\right)$$

Phase 2: Transformation from Exploration to Exploitation

Next, the algorithm transforms the state from exploration to exploitation. In this transformation, the energy of the prey is dissipated due to evading behavior. The energy level of the prey is ($E_{p}$), which is expressed as follows:(7)$$E_{p}=2E_{o}\left(1-\frac{\mathrm{iter}}{T_{m}}\right)$$

Here, $E_{0}$ is the initial state energy of the prey and $t_{m}$ is the maximum iteration. By varying the tendency of $E_{0}$, the state of the prey can be judged.

Phase 3: Exploitation

After judging the state of the prey, the Harris hawks attack the selected prey. In practice, the prey changes the evading behavior, frequently changing the attacking behavior. Four strategies are constructed in the H2ORSS algorithm for attacking prey based on evading behavior. Here, soft besiege and hard besiege are the basic strategies to attack the prey, which is decided as follows: If $\left|E_{p}\right|\geq 0.5$, then a soft besiege occurs, and if $\left|E_{p}\right|<0.5$, then a hard besiege occurs.

Soft Besiege

This attacking strategy is selected when $E_{p}\geq 0.5$ and 𝓻≥0.5 by Harris hawks. This soft besiege attacking strategy is modeled as follows:(8)$$X\left(\mathrm{iter}+1\right)=\Delta X\left(\mathrm{iter}\right)-E_{p}\left|FX_{p}\left(\mathrm{iter}-X\left(\mathrm{iter}\right)\right)\right|$$

Here, F is the jump intensity of the prey during the evading process, and it is given as $F=2\left(1-{𝓻}_{5}\right)$ and $\Delta X\left(\mathrm{iter}\right)$ represents the difference in the location vector of prey in each iteration. This difference is estimated by using the following expression:(9)$$\Delta X\left(\mathrm{iter}\right)=X_{p}\left(\mathrm{iter}\right)-X\left(\mathrm{iter}\right)$$

Hard Besiege

If $\left|E_{p}\right|<0.5$ and 𝓻≥0.5, the hard besiege strategy is selected to attack the prey. In general, these probability values show that the prey’s energy is dissipated and has low evading energy. In this case, the position of Harris hawks is updated by the following equation:(10)$$X\left(\mathrm{iter}+1\right)=X_{p}\left(\mathrm{iter}\right)-E_{p}\left|\Delta X\left(\mathrm{iter}\right)\right|$$

Soft Besiege with Progressive Rapid Dives

This strategy is selected when the prey has sufficient energy to evade form the attack. This situation is explained as $\left|E_{p}\right|\geq 0.5$ and 𝓻<0.5. Based on this behavior, the next position of the hawks is updated as follows:(11)$$Y=X_{p}\left(\mathrm{iter}\right)-E_{p}\left|FX_{p}\left(\mathrm{iter}-X\left(\mathrm{iter}\right)\right)\right|$$

As this strategy involves progressive dives, the hawk’s dive is formulated as follows:(12)$$Z=Y+B*\mathrm{lf}\left(𝒹\right)$$
where B represents the random vector; $\mathrm{lf}\left(𝒹\right)$ represents the levy flight with the dimension 𝒹. Thus, the next position is updated as follows: (13)$$\left(\mathrm{iter}+1\right)=\left\{\begin{array}{c} Y\;\;\mathrm{if}\;f\left(Y\right)<f\left(X\left(\mathrm{iter}\right)\right)\\ 𝒵\;\mathrm{if}\;f\left(𝒵\right)<f\left(X\left(\mathrm{iter}\right)\right) \end{array}\right.$$

Hard Besiege with Progressive Rapid Dives

This situation is defined as the prey has not sufficient energy to escape. This situation is formulated as $\left|E_{p}\right|<0.5$ and 𝓻<0.5. The rule for this situation is formulated as follows:(14)$$X\left(\mathrm{iter}+1\right)=\left\{\begin{array}{c} Y\;\;\mathrm{if}\;f\left(Y\right)<f\left(X\left(\mathrm{iter}\right)\right)\\ 𝒵\;\mathrm{if}\;f\left(𝒵\right)<f\left(X\left(\mathrm{iter}\right)\right) \end{array}\right.$$

Here, Y is estimated using the upcoming Equation,
(15)$$Y=X_{p}\left(\mathrm{iter}\right)-E_{p}|FX_{p}\left(\mathrm{iter}-X_{a}\left(\mathrm{iter}\right)\right)$$

Based on the above rules, the position of each hawk is updated, and the optimal solution is derived over iteration. Finally, the optimum threshold value was computed for the prediction of contrast values throughout the images.

Algorithm 1 deals with Generalized Linear Model (GLM), which is used for regression and classification tasks, is one of the key algorithms in H2O. GLM is a versatile and effective modeling approach that can deal with different data kinds and distributions.
**Algorithm 1** Pseudocode for H2OInput: PS,MaxiteOutput: Optimal ThresholdBeginInitialize → hawks population Xi (C.U.i);While (Stopping Condition Not Met) do Compute → fitness functionFor (Xi∈XPS)doUpdate →Eo and F;Update →Ep using Equation (8);End ForIf (Ep≥1)ThenUpdate position using Equation (9);End IfIf (Ep<1)ThenIf (𝓻≥0.5&&|Ep|≥0.5) Update → position using Equation (10);Else If (𝓻≥0.5&&|Ep|<0.5)ThenUpdate → position using Equation (11);Else If (𝓻<0.5&&|Ep|≥0.5)ThenUpdate → position using Equation (12);Else If (𝓻<0.5&&|Ep|<0.5)ThenUpdate → position using Equation (13);End IfEnd IfEnd WhileEnd

### 4.2. Blood Vessel Segmentation

Blood vessels are important in computing the image intensity, edges, texture, and other analyses of image features. Analyzing the diagnosis over the segmented area increases the accuracy and precision rate of any disease. Hence, the optic disk is removed from the contrast-enhanced image, and then the blood vessels are extracted using improved mask RCNN, in which ROI alignment is the first step that predicts the region of interest from the input image. In this work, pixel-wise softmax was applied for accurate segmentation of blood vessels, which was better than the CNN, RNN, RCNN, and DCNN algorithms [39,40].

OPTICS clustering stands for ordering points to identify clustering structure. It is more similar to DBSCAN clustering. OPTICS algorithm includes two measurements, which are defined as follows,
Core distance: This represents the minimum values of the radius essential to classify the given point as a core point. If the considered point is not a core point, then its core distance is indeterminate.Reachability distance: This is represented with respect to another cluster data point. The reachability distance between two points (x,y) is the highest of the core distance and then the Euclidean distance between the two points (x, y). The reachability distance is not defined if the y point is not a core point. Figure 3 represents the calculation of the reachability distance. The general procedure of M-OPTICS is defined as follows:

Next, the proposed M-OPTICS explanation is defined as follows: M-OPTICS considers three important conditions: maximum radius, distance, and number of cluster points including the core distance, core points, and reachability distance. In the M-OPTICS algorithm, the point P is known as the core point when the point is on MinPts. The reachability distance and core distance calculations are given as follows:(16)$$\mathrm{CD}\left(o\right)=\left\{\begin{array}{c} \infty \;\;\;\;\;\;\;\;\;\;\;\;\;\;\;\;\;\;\;,\;\left|o,\varepsilon \right|<MinPts\\ \mathrm{MinPts}-D\left(o\right),\;\;\;\mathrm{otherwise} \end{array}\right.$$
(17)$$R.D.\left(p,o\right)=\max\left(\mathrm{CD}\left(o\right),D\left(p,o\right)\right)$$
where p represents the object and o represents the center point. The core distance represents the lowest value, which is the radius. From the radius, the core point is different. RD represents the reachability distance, which is estimated as the highest core distance, and ε represents the radius of the data. The reachability distance data are clustered separately. The data similarities were measured by Jaccard similarity, which calculates the similarity between a finite set of samples. The calculation of the Jaccard similarity is defined as follows:(18)$$\mathrm{JD}=1-J_{\mu }\left(A,B\right)$$
(19)$$J_{\mu }\left(A,B\right)=\frac{\mu \left(A\cap B\right)}{\mu \left(A\cup B\right)}$$
where A and B represent the two points obtained from the blood vessels.

The CNN-based ensemble learning model was incorporated due to two major unique features: shared weights and local connections. The extraction of features from the input data using convolutional layers and determining the relationship between the obtained features using the pooling layers was implemented, which can be formulated as:(20)$$a_{q}^{l}=\overset{V}{\underset{p=1}{\sum }}a_{p}^{l-1}{*J}_{\mathrm{pq}}^{l}+y_{q}^{l}\;$$
where $J_{\mathrm{pq}}^{l},{\;y}_{q}^{l}$ denote the trainable parameters, and V denotes the input features. The output provided by the nonlinear layer is computed as:(21)$$x^{d}=f\left(v^{d}\right)$$
where function $f\left(v^{d}\right)$ denotes the output of the rectified linear unit. The performance of the model can be further improved by executing batch normalization. The dataset comprising of fused images of R dimensions comprised of a T number of training samples can be denoted as $H=\left\{\left(h_{i},{\mathrm{cl}}_{i}\right)|1\leq i\leq T\right\}$, where the classes are ${\mathrm{cl}}_{i}\in \mathrm{Cl}=\left\{1,2,\ldots ,M\right\}$ and the maximum count of classes is denoted as M. In the ensemble model, each model’s training is performed randomly. The input of each CNN will be $\tilde{H}=\left\{\left(\tilde{h_{i}},{\mathrm{cl}}_{i}\right)|1\leq i\leq T\right\},$ which comprises r “R feature subspaces that are randomly selected.

For instance, i and j are two identical features with dimension d, and for that similarity function ${\mathrm{sim}}_{p_{d}}^{\left(i,j\right)}$, which is computed by:(22)$${\mathrm{sim}}_{p_{d}}^{\left(i,j\right)}=\frac{{\mathrm{vector}}_{i}\times {\mathrm{vector}}_{j}}{\left\|{\mathrm{vector}}_{i}\|\times \|{\mathrm{vector}}_{j}\right\|}$$

For a different number of CNN layers and the operations involved in this study, computational complexity was evaluated, which is described as follows:(23)$$P_{v_{s_{\left(i\right)}}}=\left(1-\mu \right)\times O\left(N\right)+\mu \times O\left(N\right)=O\left(N\right)\;$$
where $O\left(N\right)$ represents the sum of iterations for performing the feature extraction and classification $\mu \;\in \left[0,1\right]$ and then $S.S._{\mathrm{upd}}$ with respect to the $f_{x}$ as follows:(24)$$S.S._{\mathrm{upd}}=\arg\left(\max_{i_{p}{\varepsilon \mathrm{idx}}_{i}}\left(f_{x}\left(P_{V_{s_{n}}}\right)\right)\right)=O\left(N\right)\;$$
where $O\left(N\right)$ represents the sum of iterations for $S.S._{\mathrm{upd}}$, which provides the near optimum feature matches from the trained set. Once the features are extracted, they are then updated by the presented method.

The output obtained from each CNN is denoted as $x=\mathrm{CNN}\;\left(\tilde{H}\right)$; the collective outputs obtained from the individual CNN are denoted as $X=\left\{x_{1},{\;x}_{2},\ldots ,{\;x}_{L}\right\}$, where L denotes the ensemble’s size, and the global output of the ensemble model is obtained by using weighted averaging of the output of the individual CNN. The weighted average of the output of the individual CNN is formulated as:$$\;\;\;G=\frac{\sum_{j=1}^{T}w_{j}s_{j}}{T}\mathrm{with}\;w_{j}\;\geq \;0$$
(25)$$\overset{T}{\underset{j=1}{\sum }}w_{j}=1$$
where $s_{j}$ denotes the score and $w_{j}$ denotes the weight of the $j-\mathrm{th}\left(j=1,\;2,\;3\right)$ model.

The classifier diversity between any two CNN models is computed as:(26)$$C_{D}\left(i,j\right)=\frac{T_{w}}{N_{T}}$$
where $N_{T}$ and $T_{w}$ denote the total number of test samples and the difference of results caused by the samples. The diversity of the ensemble model is computed as the average of the classifier diversities, which can be formulated as:(27)$$E_{D}=\frac{\sum_{i=1}^{M}\sum_{j=1}^{M}C_{D}\left(i,j\right)}{L},\;i\neq j\;$$
where $E_{D}$ denotes the diversity of the ensemble model, and $C_{D}$ denotes the classifier diversity. The classification output achieved from the weighted averaging of the individual CNN models possessed increased accuracy than the individual CNN models. Figure 4 presented the SMDTR-CNN-based land cover classification for identifying normal, DR and DME.

Table 1 addresses the ensemble deep learning model below with their filters, filter size, stride, padding, and output image size. A CNN’s fundamental building block is a convolutional layer and includes a series of filters, the parameters of which must be learned throughout the training process. The filters are often smaller in size than the real image. The pooling layer’s function is to lower the spatial size of the representation in order to reduce the number of parameters and calculations in the network; it operates independently on each feature map (channels). Maximum pooling and average pooling are the two types of pooling layers. Max pooling is a procedure commonly used for the individual CNN convolution layers listed below when they are added to a model. Maxpooling minimizes the picture dimensionality by lowering the number of pixels in the preceding convolution layer’s output. The rectified linear activation unit (ReLU) is one of the few milestones in the deep learning revolution. It is basic, but it is superior to the activation features of its predecessors such as sigmoid or tanh.

## 5. Results and Discussion

The E-CNN performance was estimated with the accuracy, precision, recall, F-score, error rate, and computational time.

### 5.1. Accuracy

Accuracy is defined as the ratio of the received input image inventive classification scheme by the assessed classification scheme, which can be formulated as:(28)$$A=\frac{T_{1}+T_{2}}{T_{1}+T_{2}+F_{1}+F_{2}}$$

From the above equation, $F_{1},F_{2}$ denote the false positive and false negative values, respectively; and $T_{1},T_{2}$ denote the true positive and true negative values, respectively. Accuracy is the significant metric for calculating the performance of the system.

### 5.2. Precision

Precision is computed by the ratio of excluding the significant classification result from the overall classification outcome. The meticulousness of the system can be measured using precision, which can be formulated as:(29)$$P=\frac{T_{2}}{T_{1}+F_{1}}\;$$

### 5.3. Recall

The recall is defined as the ratio of excluding the same classification result to the recovered results. The recall is used for measuring the comprehensiveness of the system, which can be formulated as:(30)$$R=\frac{T_{1}}{T_{1}+F_{2}}$$

### 5.4. F-Score

The F-score is computed by using the parameters of recall and precision by jointly assessing them. The results accuracy can be computed using F-score, which can be formulated as:(31)$$F_{S}=\frac{2*P*R}{P+R}$$

### 5.5. Computation Time

Computation time is the amount of time needed to complete a computational operation. Computation time is calculated by calculating the time elapsed between the classification completion and computation. The system’s efficacy is assessed in terms of computation time. It is appreciated whether the study obtained a greater accuracy with better precision of outcome in a shorter computing period.

### 5.6. Error Rate

In Table 2, the results analysis of all models is furnished in the numerical form for better understanding. The error rate is defined as the ratio of errors in the sample to the overall samples. The error rate is used to determine the system’s performance. A good system has a much lower error rate, which can be formulated as:(32)$$\mathrm{Error}\;\mathrm{Rate}=\frac{\mathrm{No}\;\mathrm{of}\;\mathrm{Errors}}{\mathrm{No}\;\mathrm{of}\;\mathrm{Samples}\;}$$

As can be seen in Figure 5, Figure 6, Figure 7, Figure 8, Figure 9 and Figure 10, we evaluated the proposed E-CNN to various state-of-the-art approaches such as SVM, KNN, enhanced CNN, and deep learning (DL). When analyzing performance, the optic disk (OD) is eliminated because it is a non-lesion area. The numerical findings suggest that our proposed E-CNN was superior. E-CNN had a mean accuracy of 99.84 percent, which was 4.38 percent greater than the benchmark. Although its effectiveness was equivalent to that of the Messidor database, it performed poorly in blood vessel segments.

Furthermore, as shown in Figure 5, Figure 6, Figure 7, Figure 8, Figure 9 and Figure 10, the accuracy, precision, recall, f-score, computational time, and error rate produced by E-CNN were much better than those acquired by other approaches. The difficulty of misclassification was exacerbated in the lesions DR and DME due to fewer samples; however, our proposed technique could still meet this obstacle. The quantitative class labels are also shown in Figure 5 to further illustrate the suggested strategy’s efficiency. In the DR lesion segmentation challenge, one can see that the E-CNN was much more accurate and robust. We also performed an ablation experiment to prove the accuracy of the proposed E-CNN. The SVM is referred to as the baseline approach for convenience. The suggested strategy has been demonstrated to generate significant improvements over the baseline regarding four targets, as shown in Figure 5, Figure 6, Figure 7, Figure 8, Figure 9 and Figure 10. The addition of preprocessing also improved the performance. The mean precision improved by 3.83 percent in comparison to the benchmark. Our proposed technology, in particular, can be simply integrated into other encoder–decoder networks, which we wish to conduct soon. Additionally, the proposed E-CNN achieved the greatest accuracy values in DR and DME diagnosis, demonstrating the efficacy of our proposed method.

In this work, ensemble convolutional neural networks (ECNNs) were used to classify images of diabetic retinopathy. A recently developed meta-heuristic method, the Harris hawks optimization (HHO) algorithm, was used to optimize the ECNN hyperparameters. Then, the Harris hawks optimization technique was used to improve the feature extraction and classification processes to obtain the most significant features. Compared to previous systems, the deep learning model provides extremely satisfactory results regarding the specificity, precision, accuracy, and recall.

## 6. Conclusions

All of the studies on the DR classification issue can be divided into two groups. The first is a binary DR diagnosis in which the individual possesses or does not. The problem with this technique is that after we realize a person has DR, we cannot tell how serious the disease is. Multi-class identification is the answer to this challenge. As previously mentioned, we classified DR into five classes or phases using multi-class classification. However, almost all of the associated studies, particularly in the early stages of DR, have been unable to appropriately define every one of the stages of DR. It is critical to identify the DR at a very early stage to treat the disease, as treating the disease at a much later date is challenging and can result in death. To our understanding, no other study has employed the IDRiR and Messidor databases to identify the milder phases of DR that we used in our study. Our approach outperformed the present advancements in detecting the mild stage. Furthermore, no one else has demonstrated the impact of a balanced dataset in previous research. The unbalanced dataset may have caused the classification accuracy to be skewed. The network can be trained on features correctly when samples in the classes are evenly distributed such as in a balanced dataset; however, in the case of asymmetrical distributions, the network performs for heavily tested classes. Furthermore, the present CNN architectures for DR identification do not consider the impact of varied hyperparameter tweaking (meta-learning) as well as its consequences. In the future, we plan to use some other deep-learning techniques for DR and DME disease classification.

Recently, CNN-based methodology has been considered to learn features for classification. However, tuning non-trainable hyperparameters for such networks is manual, intuitive, and non-trivial. In the future, a technique based on DR and DME will be proposed to adjust the CNN architecture parameters. The convolution and pooling layer number, the kernel number, and the kernel size of the convolution layer are determined by the upcoming proposed technique. Therefore, the number of untrainable hyperparameters can be reduced. There are some challenges in adapting DR and DME to a CNN. Based on the dimension of the input image, the maximum and minimum sizes of the kernel must be specified for clear classification.

## Figures and Tables

**Figure 1 diagnostics-13-01001-f001:**
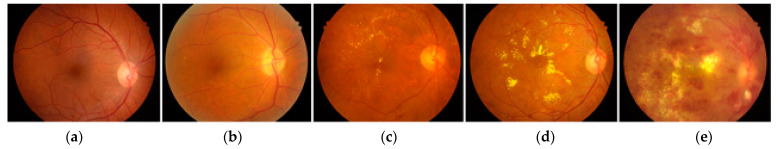
Retina images. (**a**) No DR and DME, (**b**) mild DR and DME, (**c**) moderate DR and DME, (**d**) severe DR and DME, and (**e**) non-PDR.

**Figure 2 diagnostics-13-01001-f002:**
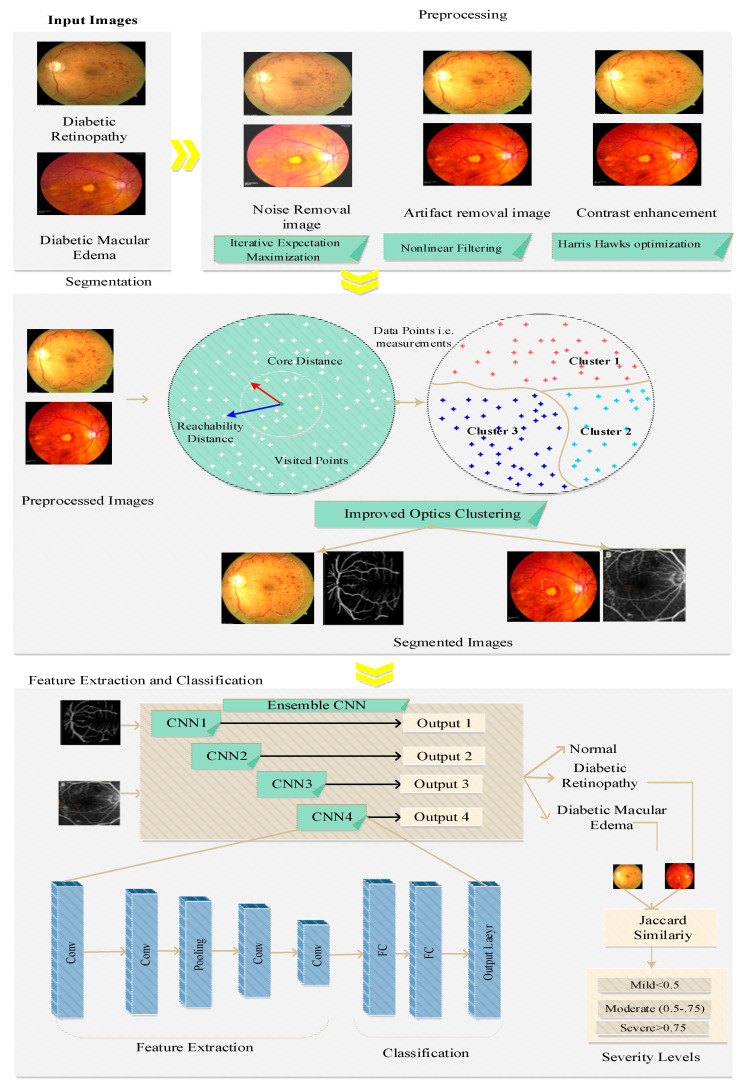
System model.

**Figure 3 diagnostics-13-01001-f003:**
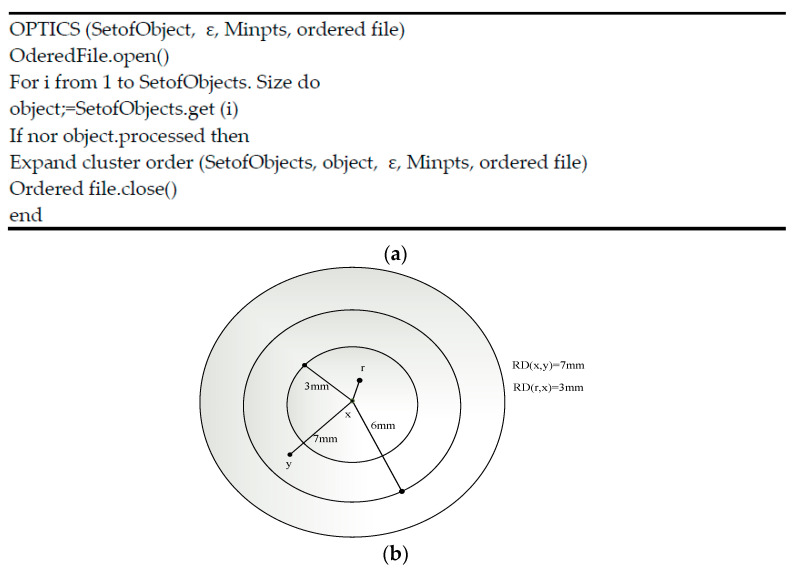
(**a**) OPTICS clustering algorithm. (**b**) Reachability distance.

**Figure 4 diagnostics-13-01001-f004:**
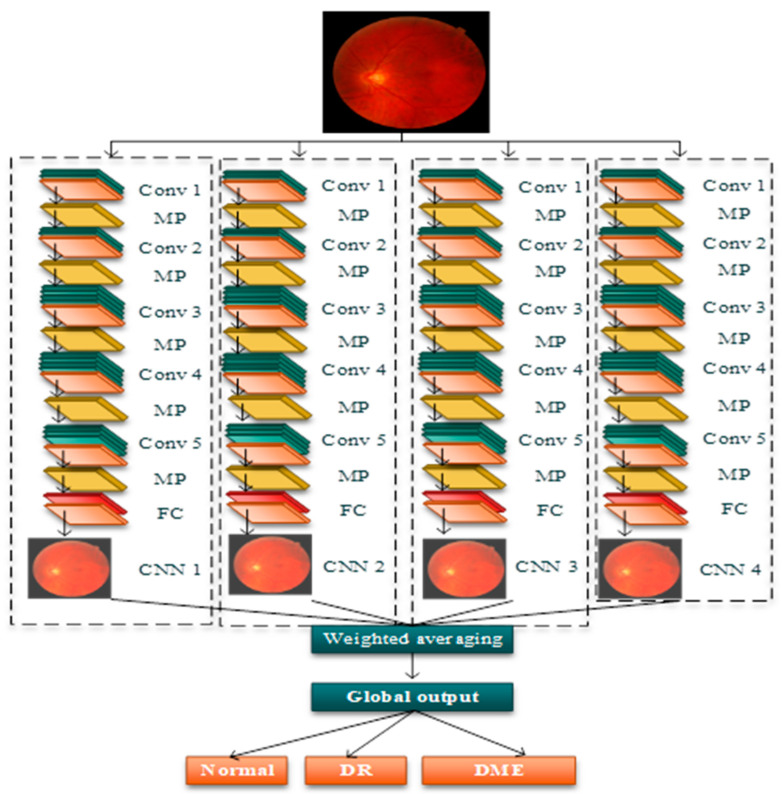
SMDTR-CNN-based land cover classification.

**Figure 5 diagnostics-13-01001-f005:**
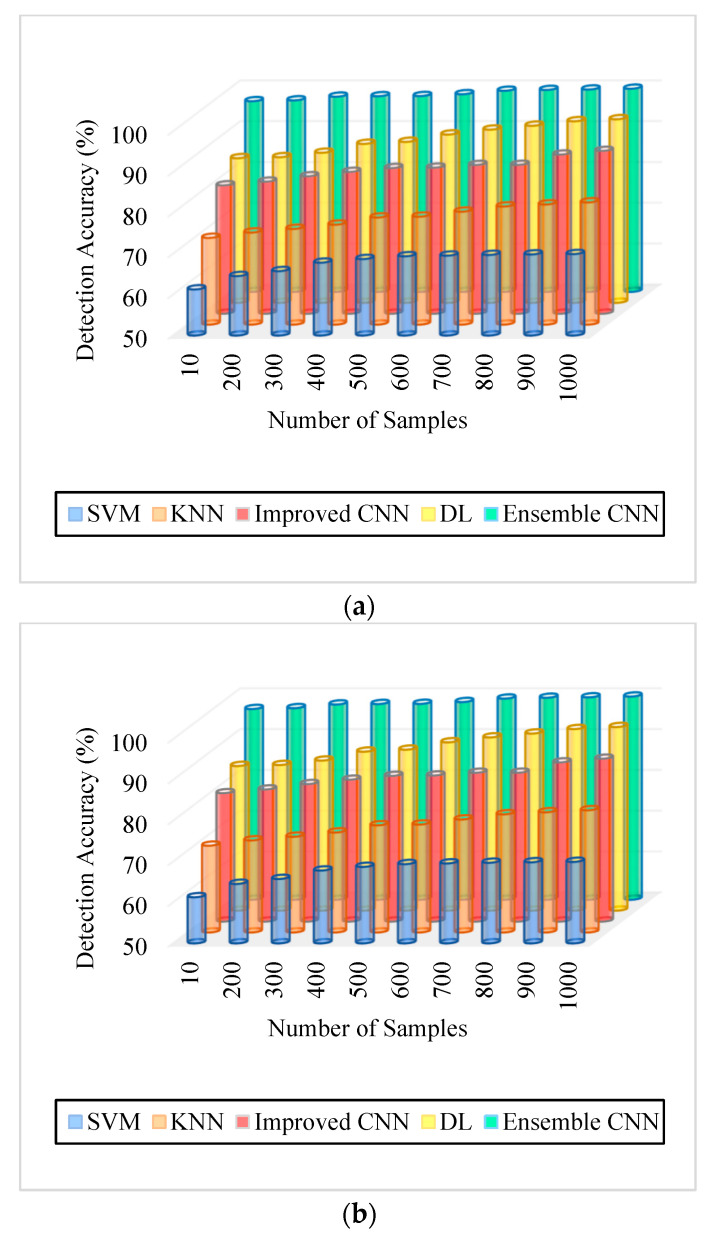
Detection accuracy (**a**). IDRiR and (**b**). Messidor.

**Figure 6 diagnostics-13-01001-f006:**
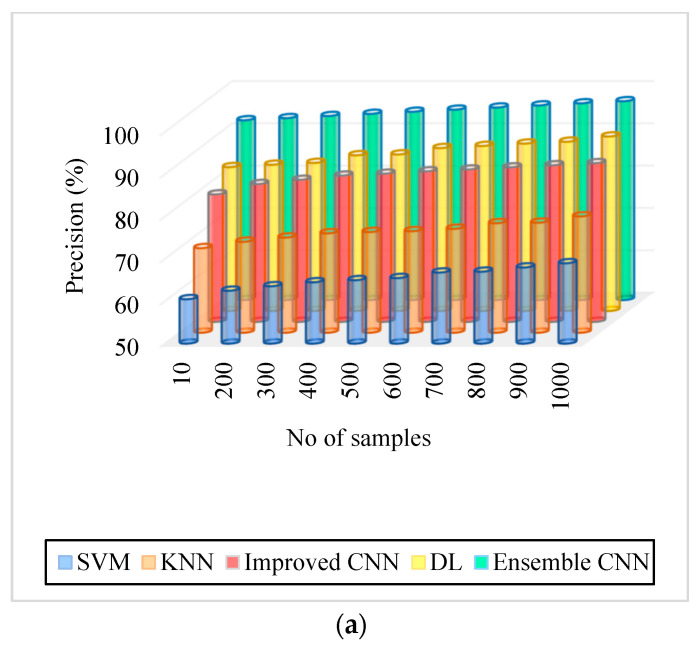
Precision. (**a**) IDRiR and (**b**) Messidor.

**Figure 7 diagnostics-13-01001-f007:**
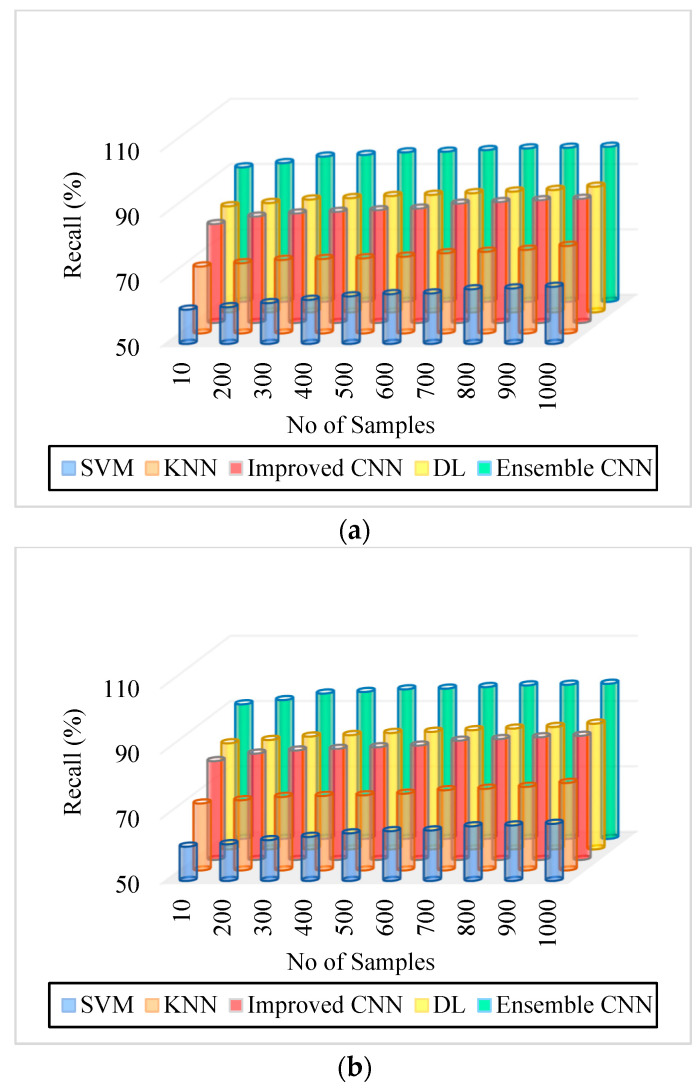
Recall. (**a**) IDRiR and (**b**) Messidor.

**Figure 8 diagnostics-13-01001-f008:**
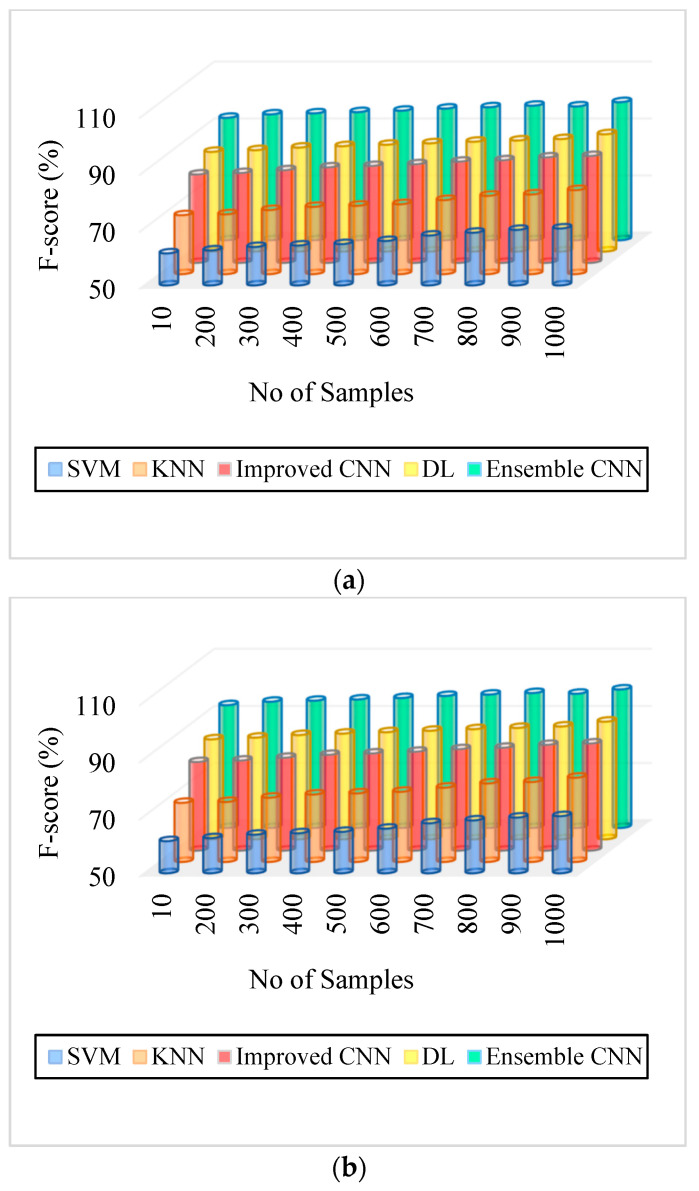
F-Score. (**a**) IDRiR and (**b**) Messidor.

**Figure 9 diagnostics-13-01001-f009:**
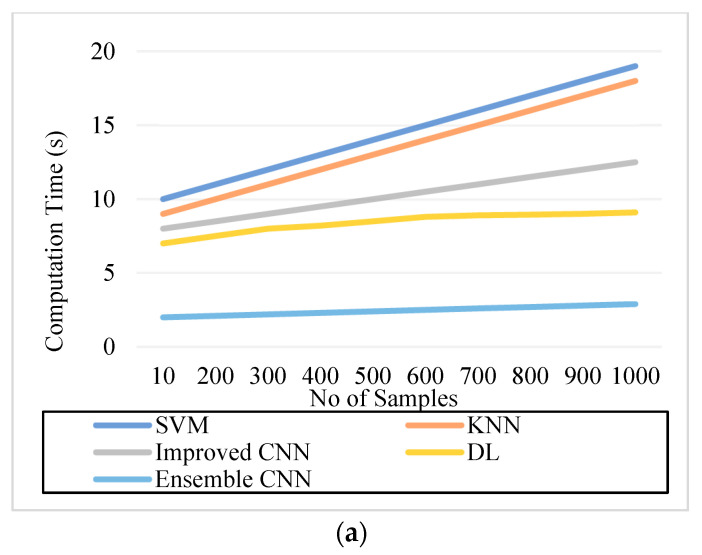
Computation time. (**a**). IDRiR and (**b**) Messidor.

**Figure 10 diagnostics-13-01001-f010:**
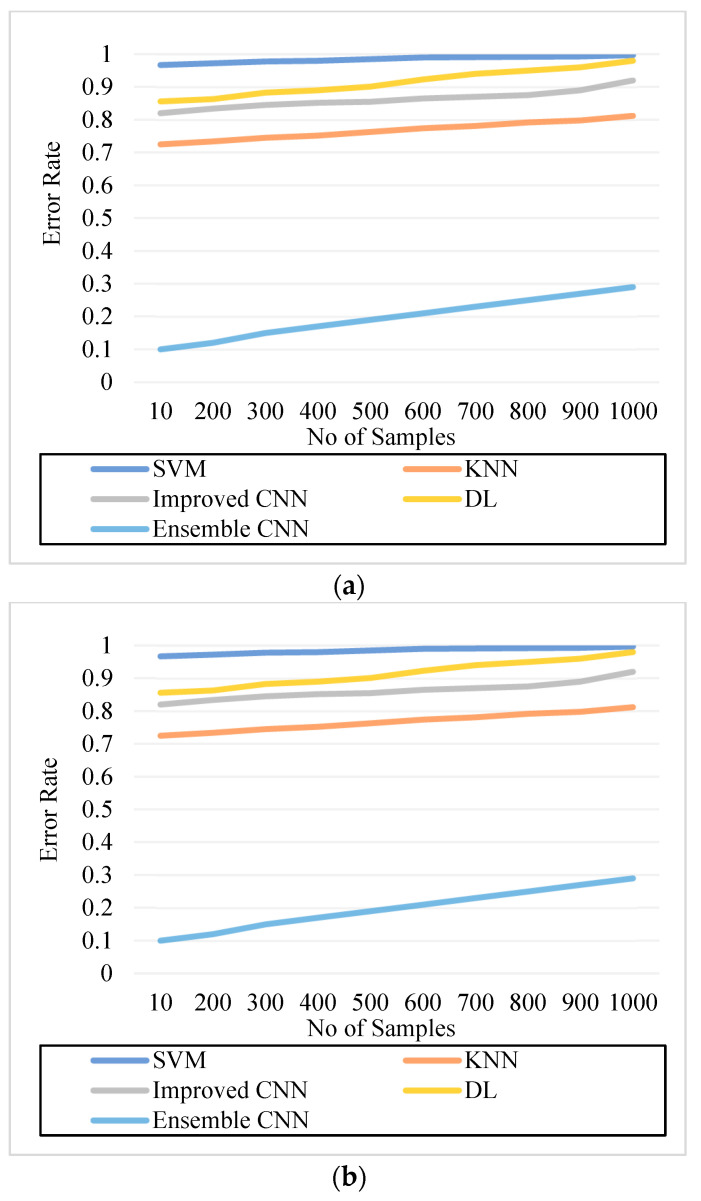
Error Rate. (**a**) IDRiR and (**b**) Messidor.

**Table 1 diagnostics-13-01001-t001:** Layers of the convolutional neural network.

Operational Layer		Filters	Filter Size	Stride	Padding	Output Image Size
Preprocessed image		-	-	-	-	224 × 224 × 3
Convolutional layer (2 times)	Convolutional	64	3 × 3 × 3	1 × 1	1 × 1	224 × 224 × 64
	ReLU	-	-	-	-	224 × 224 × 64
Pooling layer	Max pooling	1	2 × 2	2 × 2	0	112 × 112 × 64
Convolutional layer (2 times)	Convolutional	128	3 × 3 × 64	1 × 1	1 × 1	112 × 112 × 128
	ReLU	-	-	-	-	112 × 112 × 128
Pooling layer	Max pooling	1	2 × 2	2 × 2	0	56 × 56 × 128
Convolutional layer (4 times)	Convolutional	256	3 × 3 × 128	1 × 1	1 × 1	56 × 56 × 256
	ReLU	-	-	-	-	56 × 56 × 256
Pooling layer	Max pooling	1	2 × 2	2 × 2	0	28 × 28 × 256
Convolutional layer (4 times)	Convolutional	512	3 × 3 × 256	1 × 1	1 × 1	28 × 28 × 512
	ReLU	-	-	-	-	28 × 28 × 512
Pooling layer	Max pooling	1	2 × 2	2 × 2	0	14 × 14 × 512
Convolutional layer (4 times)	Convolutional	512	3 × 3 × 512	1 × 1	1 × 1	14 × 14 × 512
	ReLU	-	-	-	-	14 × 14 × 512
Pooling layer	Max pooling	1	2 × 2	2 × 2	0	7 × 7 × 512
Inner product layer	Fully connected	-	-	-	-	4096
	ReLU	-	-	-	-	4096

**Table 2 diagnostics-13-01001-t002:** Results analysis of Figure 5, Figure 6, Figure 7, Figure 8, Figure 9 and Figure 10.

Figures	SVM	KNN	Improved CNN	DL	E-CNN
Figure 5. Detection Accuracy. (a) IDRiR	61.3–69.96%	65–71%	79–83%	83–91%	94–98%
Figure 5. Detection Accuracy. (b) Messidor	54–59%	65–71%	79–84%	84–92%	96–98%
Figure 6. Precision. (a) IDRiR	60.5–69%	70–77.5%	80.12–87.5%	84–91.25%	92.5–97%
Figure 6. Precision, (b) Messidor.	60.5–69%	70–77.5%	80.12–87.5%	84–91.25%	92.5–97%
Figure 7. Recall. (a) IDRiR	60.5–67.5%	70.5–76.8%	80.25–88%	82.5–88.5%	91.2–97.5%
Figure 7. Recall. (b) Messidor.	60.5–67.5%	70.5–76.8%	80.25–88%	82.5–88.5%	91.2–97.6%
Figure 8. F-Score. (a) IDRiR	61.25–70%	70.69–79.5%	81–87.5%	85–91.2%	93–98.5%
Figure 8. F-Score. (b) Messidor	61.25–70%	70.69–79.5%	81–87.5%	85–91.2%	93–98.5%
Figure 9. Computation Time. (a) IDRiR (seconds)	11	14	10.5	8.8	2.6
Figure 9. Computation Time. (b) Messidor (seconds)	14	11	9.5	8.2	2.4
Figure 10. Error Rate. (a) IDRiR	0.985	0.774	0.865	0.923	0.15
Figure 10. Error Rate. (b) Messidor	0.99	0.792	0.865	0.923	0.19

## Data Availability

For this research work, datasets were taken from the kaggle repository site, available online at kaggle.com/datasets/mariaherrerot/idrid-dataset and https://www.kaggle.com/datasets/google-brain/messidor2-dr-grades (19 October 2022).

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
