# Peer review of "Diabetic Retinopathy and Diabetic Macular Edema Detection Using Ensemble Based Convolutional Neural Networks"

_diagnostics, 2023, doi:10.3390/diagnostics13051001_

Round 1
Reviewer 1 Report
1) the font used in equation 1, for example, but check everywhere,
is difficult to read.
2) The Figure where preprocessing is presented is very good, however the explanation of how every stage is solved should/must be improved.
3) What is IDTIR ?
4) There are more of those cases above whose meaning is given after
it first appears in the text.
5) Which is the purpose of each CNN ? Are they trained using the same set? What is either CNN learning?
6) What is the size of the training set and testing set? What are the
testing and training errors? What is the size of the set
7) Which is the function you wish to optimize using the HHO algorithm ?
8) What are the variables in 7) ?
9) What is shown in Table 1 ?
10) Figures 5 to 9 MUST be replaced by tables, so the real values can be read (instead of approximated as they are currently depicted).
11) What do you mean ? : "The effectiveness of the system is measured in terms of computation time.".
12) In Figure 10. The error rate for SVMs is almost 100% ? All methods
but Ensemble CNN show a very poor performance. Is that correct?
13) "The proposed works E-CNN is used to measure metrics for validation such as..." Could you rewrite this sentence ? ( simple idea: E-CNN performance is estimated with the following metrics... )
14) Conclusions are too short, or some kind of Section "Analysis of results" is missing.
Author Response
Reviewer-1
1) the font used in equation 1, for example, but check everywhere, is difficult to read.
Author Reaction: Thank you for your valuable comment, which will help enrich my research work.
Author Action: The expectation of log-likelihood and expected log-likelihood function can be maximized through the equations from (1) to (4). The symbol never changed anywhere; the manuscript was checked and updated.
2) The Figure where preprocessing is presented is very good, however, the explanation of how every stage is solved should/must be improved.
Author Reaction: Thank you for your excellent comment, and this increases the insight into the research work.
Author Action: All the layers' explanations have been given with suitable reasons, and the same has been updated on the manuscript page no. 10
3) What is IDTIR?
Author Reaction: Thank you for your comment
Author Action: Iterative expectation maximization (IEM) is used in the proposed Inverse Dual Tree Initial Ranging (IDTIR) approach. The IEM algorithm is an iterative approach useful for estimating statistical model parameters. The same is updated on manuscript page no. 10. Detail explanation is available in 4.1 and 4.2 on the manuscript page no. 10 - 18
4) There are more cases above whose meaning is given after it first appears in the text.
Author Reaction: Thank you for your comment
Author Action: Yes, Sir, What you are saying is correct, and I agree.
5) Which is the purpose of each CNN? Are they trained using the same set? What is either CNN learning?
Author Reaction: Thank you for your brilliant question
Author Action: Ensemble Convolutional Neural Network (E-CNN) was presented as an alternative to numerous state-of-the-art methodologies like SVM, KNN, improved Convolutional Neural Networks (CNN), and Deep Learning DL). The optic disk (OD) is excluded when assessing performance because it is a non-lesion region. The numerical results indicate that our suggested E-CNN is better. The mean accuracy of E-CNN is 99.84 percent, which is 4.38 percent higher than the benchmark. And while its effectiveness is comparable to that of the Messidor database, it performs poorly in the blood vessel sector. Furthermore, as seen in Figs 5–10, the accuracy, precision, recall, f-score, computational time, and error rate obtained by E-CNN are much higher than those obtained by other methodologies.
Purpose:
In the vast majority of applications, a single CNN model is used. There is always the possibility of employing a collection of CNN models in the same tasks as an ensemble learning strategy.
A collection of convolution neural networks. In this work, we will build an ensemble of 'N' CNN models that will be used for multi-class prediction. We will define the separate CNN models and train them sequentially. Every individual model will forecast the test data, and the final prediction of the ensemble model will be the most frequent prediction by all of the unique CNN models.
All the images were trained using Indian Diabetic Retinopathy Image Dataset (IDRID) and the Messidor database was created to aid research on computer-assisted diabetic retinopathy diagnosis. All the CNN models were trained with the same dataset in this research work.
The proposed E-CNN in the study effort learns from the Harris Hawks Optimization (HHO) approach and hyper-parameters. As a result, the number of iterations will be decreased, as will the cost. It is feasible to achieve well-optimized outcomes with little iteration.
6) What is the size of the training set and testing set? What are the testing and training errors? What is the size of the set
Author Reaction: Thank you very much for your beautiful question
Author Action: The research has provided 80% percentage for the training set and 20% for the testing set. In our proposed work, E-CNN has a mean accuracy of 99.84 percent has shown. So the error rate is 0.26 percent. The dataset size IDRID has 1488 images, and Messidor got 1748 images. In total, 3236 images were deployed.
7) Which is the function you wish to optimize using the HHO algorithm ?
Author Reaction: Thank you for your inspiring question
Author Action: In this work, Ensemble Convolutional Neural Networks (ECNN) are used to classify images of diabetic retinopathy. A recently developed meta-heuristic method, the Harris Hawks Optimization (HHO) algorithm, is used to optimize ECNN hyperparameters. Next, Harris' Hawks optimization technique is used to improve the feature extraction and classification processes to obtain the most significant features. Compared to previous systems, the deep learning model gives extremely satisfactory results in terms of specificity, precision, accuracy, and recall.
The same has been updated on the manuscript page no. 24
8) What are the variables in 7)?
Author Reaction: Thank you for your question
Author Action: In equation (7), the energy level of the prey is (Ep). Here, E0 is the initial state energy of the prey, and tm is the maximum iteration. By varying the tendency of E0, the state of the prey can be judged.
9) What is shown in Table 1?
Author Reaction: Thank you for your good question
Author Action: The layers which the Ensemble Deep learning model uses has addressed below with their filters, filter size, stride, padding, and output image size
A CNN's fundamental building block is a convolutional layer. It includes a series of filters, the parameters of which must be learned throughout the training process. The filters are often smaller in size than the real image.
The pooling layer's function is to lower the spatial size of the representation in order to reduce the number of parameters and calculations in the network; it operates independently on each feature map (channels). Maximum pooling and average pooling are the two types of pooling layers.
Max pooling is commonly used for the individual CNN convolution layers listed below when they are added to a model. Maxpooling minimizes picture dimensionality by lowering the number of pixels in the preceding convolution layer's output.
Rectified Linear Activation Unit (ReLU) is one of the few milestones in the deep learning revolution. It's basic, but it's superior to the activation features of its predecessors, such as Sigmoid or Tanh.
The same has been corrected and updated on the manuscript on page no. 18
10) Figures 5 to 9 MUST be replaced by tables, so the real values can be read (instead of approximated as they are currently depicted).
Author Reaction: Thank you for your feedback
Author Action: Figures 5 to 9 have been converted into a table according to your feedback, and the same is depicted on the manuscript page no. 24
11) What do you mean? : "The effectiveness of the system is measured in terms of computation time.".
Author Reaction: Thanks for your comment
Author Action: Computation time is the amount of time needed to complete a computational operation. Computation time is calculated by calculating the time elapsed between classification completion and computation. The system's efficacy is assessed in terms of computation time. It is appreciated if the study gets greater accuracy with better precision of outcome in a shorter computing period. It is corrected and updated on the manuscript on page no. 22
12) In Figure 10. The error rate for SVMs is almost 100%. All methods but Ensemble CNN show a very poor performance. Is that correct?
Author Reaction: Thanks for your comment
Author Action: Up to my understanding of SVM Accuracy, Precision, and Recall, F Score varies from 54% to 70%, the Computational time is 11 to 14 (Seconds), and the Error rate is 0.985% to 0.99%.
To my understanding, Accuracy, Precision, Recall, and F Score vary from 91.2% to 98.5%, Computational time ranges from 2.4 to 2.6 (seconds), and the Error rate is 0.15% to 0.91%, which is very less compared to other models.
The above-said values are mentioned in Table 2. on the manuscript page no.24
13) "The proposed works E-CNN is used to measure metrics for validation such as..." Could you rewrite this sentence? (Simple idea: E-CNN performance is estimated with the following metrics... )
Author Reaction: Thank you for your feedback
Author Action: According to the reviewer's idea, the sentence has been replaced and updated on the manuscript page no. 18
14) Conclusions are too short, or some kind of Section "Analysis of results" is missing.
Author Reaction: Thank you very much for your excellent feedback
Author Action: According to the reviewer's feedback, the conclusion section has increased. In my research work already in the 5 section, results, and discussion are depicted for result analysis. Everything is corrected and updated on the manuscript on page no. 25

Reviewer 2 Report
The paper proposed to use Ensemble Convolutional Neural Network (ECNN) to diagnose DR and DME automatically. It provided a workflow consisting of preprocessing, segmentation and CNN learning. Here is the comment. 1. The manuscript is too lengthy to follow. The total length can be reduced by one third according to the content. 2. The structure could be revised as workflow, step 1, step 2 and so on. It would be better to avoid indent of symbol identification frequently. 3. Too much contribution. Contributions are used to highlight the work briefly. However, preprocessing and segmentation are common implementing steps and the method used is not new. They could be illustrated as steps of the workflow but not the contribution of the paper. 4. Actually, results illustrated via table will be more clear than Fig 6 to 8. Or figures and tables. 5. Since the improvement is breathtaking, is there any recent algorithms to compare?Author Response
Reviewer-2
The paper proposed to use Ensemble Convolutional Neural Network (ECNN) to diagnose DR and DME automatically. It provided a workflow consisting of preprocessing, segmentation, and CNN learning. Here is the comment.
- The manuscript is too lengthy to follow. The total length can be reduced by one third according to the content.
Author Reaction: Thank you for your excellent feedback
Author Action: According to my understanding, a clear explanation is given for every segment in this research work because we have 40 references content and 32 mathematical equations explanation has given in this work. Reducing the content size will not provide a clear picture of this research work. I hope you can understand the difficulty behind it.
- The structure could be revised as workflow, step 1, step 2 and so on. It would be better to avoid indent of symbol identification frequently.
Author Reaction: Thank you for your feedback
Author Action: According to my understanding, The mathematical and analytical results only need to prove the ECCN; moreover, in this research work, harris hawks optimization methods were deployed for noise removal using Iterative Expectation Maximization, artifacts removal using non-linear filtering, and contrast enhancement. This research work is fully covered with mathematical results and analysis. So avoiding the symbols is more difficult to prove this research work. Hope you can understand me in this regard, please.
- Too much contribution. Contributions are used to highlight the work briefly. However, preprocessing and segmentation are common implementing steps, and the method used is not new. They could be illustrated as steps of the workflow but not the contribution of the paper.
Author Reaction: Thank you for your excellent feedback; it will increase the insight into our research work.
Author Action: Up to my understanding, the preprocessing and segmentation are varied from application to application; in our research, preprocessing, noise removal, artifact removal image, and contrast enhancement are available. Improved optics clustering, feature extraction, and classifications are functional in segmentation. This type of segmentation varies for applications. Everything is picturized in figure 1 step by step on the manuscript page no. 9
- Actually, results illustrated via table will be more clear than Fig 6 to 8. Or figures and tables.
Author Reaction: Thank you for your excellent feedback; as such, it is inspiring to modify the research work to bring a clear vision to the readers.
Author Action: According to the reviewer feedback, fig 6 to fig 8 were converted as a table, and the same is updated on the manuscript page no. 24
- Since the improvement is breathtaking, are there any recent algorithms to compare?
Author Reaction: Thank you for your beautiful feedback;
Author Action: Yes, other models were compared with this work
Support Vector Machine (SVM), K Nearest Neighbourhood (KNN), Improved Convolutional Neural Network, and Deep Learning (DL) were compared with our proposed Ensemble Convolutional Neural Network (ECNN) in terms of Accuracy, Precision, Recall, FScore, Computational time and Error rate, in all segments ECNN our proposed model working fine in all the factors. The same is deployed in Table 2 and updated on the manuscript page no. 24

Reviewer 3 Report
It seems that the paper presents a solid and interesting method. The results show that the proposed process outperforms the state-of-the-art stated methods.
Author Response
Reviewer-3
It seems that the paper presents a solid and interesting method. The results show that the proposed process outperforms the state-of-the-art stated methods.
Author Reaction: Thank you very much for your feedback and acceptance of our research work.
Round 2
Reviewer 1 Report
The new version really means a great improvement thus I agree
with the paper in its current form and I do not have further comments.